# Explaining the increment in coronary heart disease mortality in Mexico between 2000 and 2012

Carmen Arroyo-Quiroz [1,2], Martin O'Flaherty[3], Maria Guzman-Castillo[4], Simon Capewell[3], Eduardo Chuquiure-Valenzuela[5], Carlos Jerjes-Sanchez[6], Tonatiuh Barrientos-Gutierrez[1]*

1 Center for Research on Population Health, National Institute of Public Health, Cuernavaca, Mexico, 2 Universidad Autonoma Metropolitana- Unidad Lerma, Lerma de Villada, Mexico, 3 Institute of Psychology, Health and Society, University of Liverpool, Liverpool, United Kingdom, 4 Population Research Unit, Faculty of Social Sciences, University of Helsinki, Helsinki, Finland, 5 National Institute of Cardiology, Mexico City, Mexico, 6 Escuela de Medicina y Ciencias de la Salud, Instituto Tecnológico de Monterrey, Instituto de Cardiología y Medicina Vascular, TecSalud, Monterrey, Mexico

* tbarrientos@insp.mx

## Abstract

### Background

Mexico is still in the growing phase of the epidemic of coronary heart disease (CHD), with mortality increasing by 48% since 1980. However, no studies have analyzed the drivers of these trends. We aimed to model CHD deaths between 2000 and 2012 in Mexico and to quantify the proportion of the mortality change attributable to advances in medical treatments and to changes in population-wide cardiovascular risk factors.

### Methods

We performed a retrospective analysis using the previously validated IMPACT model to explain observed changes in CHD mortality in Mexican adults. The model integrates nationwide data at two-time points (2000 and 2012) to quantify the effects on CHD mortality attributable to changes in risk factors and therapeutic trends.

### Results

From 2000 to 2012, CHD mortality rates increased by 33.8% in men and by 22.8% in women. The IMPACT model explained 71% of the CHD mortality increase. Most of the mortality increases could be attributed to increases in population risk factors, such as diabetes (43%), physical inactivity (28%) and total cholesterol (24%). Improvements in medical and surgical treatments together prevented or postponed 40.3% of deaths; 10% was attributable to improvements in secondary prevention treatments following MI, while 5.3% to community heart failure treatments.

**Data Availability Statement:** The datasets supporting the conclusions of this article are

available at: (1) Population size: Mexican National Population Council (CONAPO). https://www.inegi. org.mx/app/tabulados/interactivos/?px=Poblacion_ 01&bd=Poblacion#variables (2) Mortality: Mexican Ministry of Health Information System http://www. dgis.salud.gob.mx/contenidos/basesdedatos/bdc_ defunciones_gobmx.html (3) Number of patients: Mexican Ministry of Health Information System [29] http://www.dgis.salud.gob.mx/contenidos/ basesdedatos/bdc_egresoshosp_gobmx.html (4) Treatments: Mexican Ministry of Health and National Registry of Acute Coronary Syndromes (RENASICA). Data is only available upon request from the RENASICA Executive Committee: http:// www.renasica.mx/contacto.html (5) Risk factors: National Nutrition and Health Surveys (ENSANut) https://ensanut.insp.mx/ In the case National Registry of Acute Coronary Syndromes, the data are available upon reasonable request from the RENASICA Executive Committee. The authors confirm that they had no special access privileges in accessing these data sets. Contact information: Dr. Carlos Martínez (RENASICA Executive Committee), email: martinez@renasica.mx, info@renasica.mx and publicaciones@renasica.mx The rest of the datasets are publicly available and anonymous.

**Funding:** Carmen Arroyo-Quiroz received support from CONACyT's Scholarship Program for Doctoral Studies. Tonatiuh Barrientos-Gutierrez received support from Harvard University through the Lown Scholar's program (https://www.hsph.harvard.edu/ lownscholars/scholars/). The funders had no role in the study design or the analysis and interpretation of the data. All authors and their institutions reserve intellectual freedom from the funders.

**Competing interests:** The authors have declared that no competing interests exist.

## Conclusions

CHD mortality in Mexico is increasing due to adverse trends in major risk factors and suboptimal use of CHD treatments. Population-level interventions to reduce CHD risk factors are urgently needed, along with increased access and equitable distribution of therapies.

## Introduction

Coronary heart disease (CHD) remains as one of the most important worldwide mortality causes. Currently, most developed countries are experiencing a decrease in CHD mortality, and some have reduced their rates by 50%, compared to the 1970s [1]. The reasons for these decreases are variable, yet, they have been attributed to declines in major risk factors and advances in medical and surgical treatment [2, 3]. In low and middle-income countries, the behavior of the CHD epidemic differs, while CHD mortality in some countries is decreasing, many other countries are still experiencing increases in CHD mortality [4]. Mexico, an upper middle-income country with 129 million inhabitants, is one of the countries that are still in the rising phase of the CHD epidemic [5].

In the past century, Mexico was characterized as a country of low CHD mortality. In 1980, the age-standardized mortality rate per 100,000 inhabitants were 55.6 and 33.8 for men and women, respectively [6]. However, between 1980 and 2010, substantial changes in the country led to a 48% increase in CHD mortality [7]. An explosive increase in obesity and diabetes has been proposed as the main factor for CHD mortality increase [8]. Simultaneously, the country experienced a reduction in smoking rates, from 25.9% in 1980 to 10% in 2012 [9], which should have led to a reduction in coronary morbidity and mortality. Also, since 2004, access to CHD clinical and surgical treatments increased under *Seguro Popular*, a health insurance program to provide access to packages of health services for people without access to social security services [10, 11]. Treatments such as thrombolysis, coronary-artery bypass grafting (CABG), coronary angioplasty, ACE inhibitors and statins are now more commonly used and available for more people [12, 13]. While the rises in CHD mortality in Mexico are clear, we still lack an analysis of the fundamental causes of these changes [14].

In the last decades, various models have been developed to measure the contribution of risk factors and treatment changes in CHD mortality [2, 3, 15, 16]. These models use up-to-date scientific evidence to estimate the contribution of population changes in risk factors and consider the accessibility and effectiveness medical and surgical treatments, to changes in CHD mortality [17]. The IMPACT model has been implemented in United Kingdom, the United States, New Zealand, Syria, Tunisia and China among others [3, 16, 18–21]. IMPACT studies in high-income countries concluded that CHD mortality reductions were largely attributable to improvements in risk factors, such as obesity and hypertension, which explained from 44% of the reduction in the United States [3] to 76% in Finland [22]. IMPACT has also been implemented in low and middle-income countries that were experiencing increases in CHD mortality rates, such as Tunisia [18] and Syria [23], where risk factors explained more than 60% of the change.

To our knowledge, no study has estimated the contribution of risk factors and medical and surgical treatments on CHD mortality in Mexico. This information is key to identify potential targets for public and medical policy aiming to reduce the burden of CHD. We aimed quantify what proportion of the CHD mortality change between 2000 and 2012 that is attributable to advances in medical and surgical treatment and changes in population-wide cardiovascular risk factors using the IMPACT model.

## Material and methods

The IMPACT policy model was used in this study to quantify the effects on CHD mortality attributable to variations in each population risk factors and treatment modalities between 2000 and 2012 [2, 17, 23–26]. The model methodology has been described in detail elsewhere [3, 16, 17, 23]. Briefly, the IMPACT model is used to estimate the number of coronary heart disease (CHD) death change attributable to changes in specific cardiac interventions, treatments, or risk factors. In this study, adult data including: (1) number of CHD patients, (2) use of specific medical and surgical treatments, (3) effectiveness of specific treatments for CHD, (4) population trends of major cardiovascular risk factors (smoking, total cholesterol, hypertension, obesity, and diabetes), were incorporated into the model [2, 3, 16, 25].

### Data sources

National information on mortality, morbidity, hospital discharges, medical and surgical treatments, and cardiovascular risk factors was obtained for the years 2000 and 2012. All data are national and grouped by age and sex. Data used are described de in detail in S1 Appendix, briefly, we used four data sources:

- Population size: Mexican National Population Council (CONAPO) [27]

- Mortality: Mexican Ministry of Health Information System [7, 28]

- Number of patients: Mexican Ministry of Health Information System [29]

- Treatments: Mexican Ministry of Health and National Registry of Acute Coronary Syndromes (RENASICA) [13, 30, 31]

- Risk factors: National Nutrition and Health Surveys (ENSANut) [32, 33]

In the case National Registry of Acute Coronary Syndromes, the data are available only upon request from the RENASICA Executive Committee, the rest of the datasets are publicly available and anonymous. We limited our CHD mortality analysis to the 2000 to 2012 period, since 2012–2013 was the last wave of RENASICA data.

### Deaths prevented or postponed (DPP)

The primary output was Deaths Prevented of Postponed (DPP) for CHD. DPP represents the difference between the 2012 expected CHD deaths, calculated assuming no change in the distribution of risk factors and medical and surgical treatments available in 2000, to the CHD mortality observed in 2012. Mortality rates from CHD were calculated using the underlying cause of death: International Classification of Diseases (ICD) -10 codes I20-I25 [3]. We used demographic data obtained from the Mexican National Population Council (CONAPO) and mortality data for adults aged 25 years and older from the Health Information System from the Mexican Health Ministry to calculate the CHD age- sex-group-specific mortality rates in 2000 and 2012. The expected number of CHD deaths in 2012 was calculated by multiplying age-sex group-specific mortality rates in 2000 by the corresponding population size of each 10-year age-sex stratum in 2012 [2, 3, 16, 25, 26]. A positive DPP implies a decrease in observed mortality, relative to the expected mortality, while a negative DPP implies an increase in observed relative to expected mortality.

$$DPP = Expected\ mortality_{2012} - Observed\ mortality_{2012}$$

The obtained DPP is the number of deaths to be explained by the model, this was achieved thorough the contribution of $DPP_{treatment}$, which represents de DPP that is attributable to

changes in medical and surgical treatments, and $DPP_{risk}$, that represents the DPP attributable to changes in risk factors.

$$DPP = Expected\ mortality_{2012} - Observed\ mortality_{2012} = DPP_{treatment} + DPP_{risk} + e$$

Where $e$ represents an error term that captures the change that is not explained by our model. In the next sections we will explain how every DPP was calculated.

## Mortality changes attributable to treatment uptake

The first step in the estimation of DPP is to calculate $DPP_{treatment}$, which is a combination of the individual $DPP_{treatment}$ as a result of each intervention/ therapy in each group of patients in 2012, stratified by age and sex. To achieve this, we specified relevant treatments for each of the nine mutually exclusive patient groups [3, 17, 34, 35]:

- Patients treated in hospital for myocardial infarction (ST-elevation myocardial infarction and non-ST elevation acute coronary syndrome)

- Patients admitted to the hospital with unstable angina

- Community-dwelling patients who have survived a myocardial infarction (MI) for over a year

- Patients who have undergone a revascularization procedure up to and including the years 1980 and 2012: Coronary artery bypass grafting (CABG), or a percutaneous coronary intervention (PCI)

- Community-dwelling patients with stable coronary artery disease

- Patients admitted to hospital with heart failure (due to CHD)

- Community-dwelling patients with heart failure (due to CHD)

- Hypercholesterolemic subjects without CHD eligible for cholesterol-lowering therapy such as statins

- Hypertensive individuals without CHD eligible for anti-hypertensive therapy

To obtain the $DPP_{treatment}$ for an specific group of patients and therapy we used the number of people in each diagnostic group of patients in 2000 and then it was multiplied by the proportion of patients who received a particular treatment, by their case fatality rate over a 1-year period, and by the relative reduction in the 1-year case fatality rate reported for that treatment in the largest and most recent meta-analysis [3, 17, 36, 37] (S2 Appendix).

For example, in 2012, 10,752 men aged 45 to 54 were hospitalized with myocardial infarction (MI). The expected age-specific 1-year case-fatality rate was 5.4%. 79% were prescribed acetylsalicylic acid, with an expected mortality reduction of 15%. The number of DPP was then calculated as:

$$DPP_{MI-ASA} = 10\ 752 \times 79\% \times 15\% \times 5.4\% = 69\ deaths\ prevented\ or\ postponed$$

This process was replicated for every sex-age group, patient group, and therapy. Some special considerations were made to these initial calculations. We assumed that the proportion of treated patients actually taking therapeutically effective levels of medication (adherence), was 100% among hospitalized patients, 70% among symptomatic patients in the community, and 50% among asymptomatic patients in the community [3, 25].

In the case of individual patients that were receiving multiple treatments, we applied the Mant and Hicks cumulative-relative benefit approach to estimate the potential effect on the

relative decrease in the case fatality rate for those patients [3, 36–39]

$$Relative\ Benefit = 1 - [(1 - relative\ reduction\ in\ case\ fatality\ rate\ for\ treatment\ A)$$
$$\times (1 - relative\ reduction\ in\ case\ fatality\ rate\ B) \times \ldots\ldots$$

Potential overlaps between different groups of patients were detected and adjustments were made to prevent double-counting (e.g., 50% of patients having CABG surgery had previous myocardial infarction) (S3 Appendix) [3, 17, 25]. Briefly, we subtracted the DPPs calculated in the treatment component from the DPPs calculated in the risk factors component. Additional assumptions are listed in S3 Appendix.

After carrying out all these calculations, we combined the $DPP_{treatment}$ for every patient group and treatment until we obtained a single $DPP_{treatment}$ by age-sex group that considered all groups of patients and possible therapies.

## Mortality changes attributable to risk factor changes

The second component of the IMPACT model includes estimating the number of $DPP_{risk}$ for CHD due to changes in the cardiovascular risk factors for every age-sex group. We included six major cardiovascular risk factors in the model: smoking, physical inactivity, body mass index (BMI), systolic blood pressure, total serum cholesterol, and diagnosed diabetes [2, 3, 25]. DPPs associated to an absolute change in each risk factor between 2000 and 2012 were calculated using: a) a regression-based approach for factors measured on a continuous scale (such as total blood cholesterol, systolic blood pressure and BMI) or b) a population-attributable risk fraction (PARF) approach to estimate the effect of variations in categorical variables.

In the case of the regression-based approach, we used sex and age-specific independent regression coefficients of mortality benefit for a unit change in the mean of each risk factor [2, 3, 25, 40]. In S4 Appendix we listed the sources for the regression (beta) coefficients utilized in these analyses. We estimated the number of $DPP_{risk}$ as a result of the change in the mean value of each of these risk factors considering the product of: the number of deaths from CHD in 2000 (the baseline year), the subsequent change in that risk factor, and the regression coefficient measuring the variation in mortality from CHD per unit of absolute change in the risk factor [3, 16, 17].

$$DPP_{risk} = (1 - (EXP(coefficient * change)) * deaths\ in\ 2000$$

For example, there were 4,069 CHD deaths in 12,629,000 men aged 65–74 Years in 2000. In this groups, mean systolic blood pressure reduced by 0.82 mmHg (from 130.4 in 2000 to 129.6 mmHg in 2012) [3, 20, 36, 37]. Previous meta-analyses reported an expected age- and sex-specific decline in mortality of 50% for every 20 mmHg. decrease, generating a logarithmic coefficient of –0.035 [3, 20, 36, 37]. The number of DPP was then estimated as:

$$DPP_{hypertension} = (1 - (EXP(coefficient * change)) * deaths\ in\ 2000$$

$$DPP_{hypertension} = (1 - EXP(-0.035 * 0.82)) * 4069 = 115$$

We repeated this process for every age-sex group and continuous risk factor considered.

We applied the population-attributable risk fraction (PARF) approach to estimate the effect of variations in categorical variables (prevalence of smoking, diabetes, and physical inactivity) [2, 3, 20, 36]. Sources for the relative risks (RR) utilized in these analyses are listed in the

S4 Appendix. To estimate the PARF, we applied the following formula [2, 3, 25]:

$$PARF = \frac{P \times (RR - 1)}{1 + P \times (RR - 1)}$$

Where P is the risk factor prevalence, and RR is a relative risk. We then estimated DPP as the CHD deaths in 2000 multiplied by the difference in the PARF during the period (2000–2012).

$$DPP_{risk} = CHDdeaths_{2000} * (PARF_{2012} - PARF_{2000})$$

For example, suppose that the prevalence of diabetes among men aged 65–74 years was 14.5% in 2000 and 20.7% in 2012. Assuming a RR = 1.93, the PARF in 2000 was 0.119 and 0.161 in 2010. The number of CHD deaths is 2000 was 123,055. The DPP attributable to the change in diabetes prevalence was therefore:

$$DPP_{diabetes} = (123,055)*(0.161 - 0.119) = 5,168$$

This calculation was then repeated for each sex-age group and for every categorical risk factor. Finally, we obtained a total $DPP_{risk}$ for each sex-age group by adding the DPP obtained for each risk factor across all risk factors. As independent coefficients of regression and relative risks were derived from multivariate analyses for each risk factor, we assumed that there was no further overlap across all risk factors considered [2, 3, 25].

## Uncertainty analysis

Using Monte Carlo simulation, we computed 95% uncertainty intervals around the model output [2, 3, 25]. To obtain these calculations, we replaced all fixed input parameters used in the model by suitable probability distributions and then we repeatedly recalculate the model output with values sampled from the given input distributions (S5 Appendix) [2, 3, 25]. We used the Excel add-in Ersatz software (www.epigear.com) to do 1,000 runs to determine the 95% uncertainty intervals of the DPP (2.5th and 97.5th centile values corresponding to the lower and upper limits) [2, 3, 25].

## Results

In Mexico between 2000 and 2012, CHD crude mortality rates increased by 33.8% in men and by 22.8% in women (from 105 to 140 and from 81 to 100 per 100,000 men and women, respectively). In 2012, we observed an excess of 9,370 CHD deaths, compared to those expected from baseline mortality rates in 2000 (Table 1).

An excess of approximately 10,580 CHD deaths was attributable to changes in major cardiovascular risk factors (UI -12,273; -9,213 Table 2). Improvements in medical and surgical treatments together prevented or postponed approximately 3,900 deaths by 2012 (UI 1829; 5950; Table 3). After subtracting the prevented or postponed deaths from the excess of deaths related to risk factors, an increase of 6624 deaths were obtained, which represent 71% of the total CHD mortality rise in the study period. The biggest contributor to CHD mortality was the increase in diabetes prevalence (from 7.7% to 10.7%), which led to an estimated 3,565 additional CHD deaths (UI 2864; 4271) (Fig 1). The second-largest contribution came from physical inactivity (from a prevalence of 9% to 19%), which led to an estimated 3,395 additional deaths (UI 2,775; 4,499). Increases in total cholesterol, mean BMI, and systolic blood pressure resulted in an estimated additional 2219, 1699 and 1134 deaths, respectively. The only risk factor that improved was the prevalence of smoking, which decreased by 0.03 percent points and

**Table 1. Population sizes and death rates due to CHD in Mexico, 2000 and 2012.**

| Sex and age group | | 2000 | | | 2012 | | | Adjusted to 2000 rate | Diference observed-expected |
|---|---|---|---|---|---|---|---|---|---|
| | | Population | CHD deaths | Crude rate per 100,000 | Population | CHD deaths | Crude rate per 100,000 | | |
| Male | 25–34 | 8'063,423 | 351 | 4.4 | 8,813,802 | 668 | 7.6 | 383.7 | 284 |
| | 25–44 | 5'812,452 | 1006 | 17.3 | 7,651,545 | 1623 | 21.2 | 1324.3 | 299 |
| | 45–54 | 3'861,354 | 2246 | 58.2 | 5,682,366 | 3660 | 64.4 | 3305.2 | 355 |
| | 55–64 | 2'490,327 | 4069 | 163.4 | 3,699,270 | 6320 | 170.8 | 6044.3 | 276 |
| | 65–74 | 1'556,271 | 5857 | 376.3 | 2,105,313 | 8930 | 424.2 | 7923.3 | 1007 |
| | 75–84 | 691,352 | 5879 | 850.4 | 1,021,032 | 10578 | 1,036.0 | 8682.5 | 1896 |
| | 85+ | 208,240 | 4480 | 2151.4 | 322,970 | 9500 | 2,941.4 | 6948.3 | 2552 |
| Female | 25–34 | 8'496,491 | 137 | 1.6 | 9,657,083 | 186 | 1.9 | 155.7 | 30 |
| | 25–44 | 6'178,879 | 379 | 6.1 | 8,463,679 | 560 | 6.6 | 519.1 | 41 |
| | 45–54 | 4'103,677 | 926 | 22.6 | 6,287,850 | 1401 | 22.3 | 1418.9 | -18 |
| | 55–64 | 2'669,170 | 2279 | 85.4 | 4,095,052 | 2943 | 71.9 | 3496.5 | -553 |
| | 65–74 | 1'687,604 | 4257 | 252.3 | 2,390,205 | 5662 | 236.9 | 6029.3 | -367 |
| | 75–84 | 800,691 | 5583 | 697.3 | 1,223,432 | 9443 | 771.8 | 8530.7 | 912 |
| | 85+ | 279,845 | 6290 | 2247.7 | 441,089 | 12572 | 2,850.2 | 9914.2 | 2658 |
| | | GAP TO EXPLAIN | | | | | | | 9370 |

prevented 651 deaths (UI 102; 1486). Some risk differences between men and women are important; women had larger increases in diabetes, smoking and cholesterol, while men had larger increases in systolic blood pressure and physical inactivity. Although women experienced a larger increase in smoking prevalence, the prevalence for men remained higher.

Medical and surgical treatments together prevented or postponed approximately 3,950 deaths by 2012 (UI 1829; 5950). The largest mortality reductions came from secondary prevention treatments following MI, which prevented or postponed 1,045 deaths (11.2%), mostly due to statin use increases (Table 3). Approximately 820 deaths (5.6%) were prevented by improvements in the treatment of heart failure in the community (particularly acetylsalicylic acid and spironolactone), and 605 (6.4%) were attributable to primary prevention (statins and antihypertensives). Treatment of angina pectoris in the community prevented 532 deaths (5.7%), largely attributable to revascularization, which prevented 3.8% deaths as compared with deaths in the year 2000. Improvements in acute phase management (MI and unstable angina) were modest and prevented approximately 306 deaths (3.3%)

The relative contribution of new therapies and improvements in the risk factor to the overall decrease in CHD deaths in 2012 was consistent through sensitivity analyses (Fig 1). The largest part of the mortality increase was explained by large rises in diabetes, physical inactivity, and total cholesterol. Likewise, mortality reductions were linked to lower smoking prevalence and an increase in therapies for secondary prevention following MI and heart failure treatment in the community.

**Table 2. Deaths from coronary heart disease that were prevented or postponed as a result of changes in population risk factors in Mexico from 2000 to 2012.**

| Risk factor | Risk factor level | | Risk factor change | | Deaths prevented or postponed | | | | | |
|---|---|---|---|---|---|---|---|---|---|---|
| | 2000 | 2012 | Absolute | Relative | Best estimate | Minimun estimate | Maximum estimate | Best estimate | Minimun estimate | Maximum estimate |
| | | | | | No. Deaths | | | % of total change | | |
| *SYSTOLIC BLOOD PRESSURE (mmHg)* | | | | | | | | | | |
| All | 124.1 | 123.2 | 0.94 | 0.01 | - 1,137.3 | - 1,686.4 | - 704.8 | 12.1% | 7.5% | 18.3% |
| Men | 125.3 | 124.6 | 0.71 | 0.01 | - 1,251.8 | - 1,736.8 | - 857.5 | 13.4% | 9.2% | 18.4% |
| Women | 123.0 | 121.9 | 1.14 | 0.01 | 114.5 | 77.0 | 156.2 | -1.2% | -1.7% | -0.8% |
| *CHOLESTEROL (mmol/L)* | | | | | | | | | | |
| All | 4.9 | 5.1 | (0.19) | (0.04) | - 2,203.0 | - 3,942.7 | - 907.6 | 24% | 10% | 43% |
| Men | 4.9 | 4.9 | - | - | - | - | - | 0% | 0% | 0% |
| Women | 5.0 | 5.3 | (0.35) | (0.07) | - 2,203.0 | - 3,837.0 | - 626.8 | 24% | 7% | 40% |
| *BMI (Kg/ m2)* | | | | | | | | | | |
| All | 26.4 | 28.6 | (2.24) | (0.08) | - 1,692.6 | - 2,086.5 | - 1,261.5 | 18% | 13% | 23% |
| Men | 25.8 | 28.0 | (2.23) | (0.09) | - 944.0 | - 1,228.6 | - 666.9 | 10% | 7% | 13% |
| Women | 27.0 | 29.2 | (2.24) | (0.08) | - 748.5 | - 1,085.8 | - 415.8 | 8% | 4% | 12% |
| *SMOKING (%)* | | | | | | | | | | |
| All | 0.22 | 0.19 | 0.03 | 0.10 | 649.6 | - 102.6 | 1,482.6 | -7% | -16% | 1% |
| Men | 0.35 | 0.30 | 0.04 | 0.13 | 532.6 | 9.0 | 988.0 | -6% | -10% | 0% |
| Women | 0.11 | 0.10 | 0.01 | 0.07 | 116.9 | - 427.8 | 643.3 | -1% | -7% | 5% |
| *PHYSICAL INACTIVITY (%)* | | | | | | | | | | |
| All | 0.11 | 0.18 | (0.08) | (0.75) | - 2,609.1 | - 3,129.3 | - 2,045.2 | 28% | 22% | 33% |
| Men | 0.10 | 0.19 | (0.09) | (0.95) | - 1,500.5 | - 1,852.7 | - 1,140.9 | 16% | 12% | 20% |
| Women | 0.11 | 0.18 | (0.06) | (0.58) | - 1,108.7 | - 1,473.2 | - 637.1 | 12% | 7% | 16% |
| *DIABETES (%)* | | | | | | | | | | |
| All | 0.06 | 0.09 | (0.04) | (0.58) | - 4,041.6 | - 4,842.9 | - 3,247.6 | 43% | 35% | 51% |
| Men | 0.05 | 0.09 | (0.03) | (0.48) | - 1,681.1 | - 1,270.1 | - 2,301.8 | 18% | 14% | 25% |
| Women | 0.06 | 0.10 | (0.04) | (0.67) | - 2,360.5 | - 1,761.5 | - 2,976.0 | 25% | 19% | 32% |
| **TOTAL RISK FACTORS** | | | | | **-11,034.1** | **- 12,796.9** | **- 9,605.7** | **118%** | **102%** | **138%** |

## Discussion

We aimed to estimate the contribution of changes in risk factors and treatments to CHD mortality increases in Mexico. We found that CHD crude mortality rates increased substantially between 2000 and 2012 (33.8% in men and 22.8% in women). This mortality increase was attributable to adverse trends in major risk factors, mainly diabetes, cholesterol, and physical inactivity. Mortality rises were mitigated by medical interventions, which prevented or postponed approximately 3900 deaths, potentially decreasing overall CHD mortality by about 40%.

Most of the previous IMPACT models were implemented in high income countries that are experiencing a decrease in CHD mortality, mostly attributable to risk factor reductions [3, 17, 20, 36, 40]. Those are the cases of United Kingdom, Denmark, Japan, Netherlands or the United States, among others [3, 20, 25, 35, 40]. In Latin America, only Argentina has implemented the IMPACT model and, while they have experienced some increases in diabetes and obesity, the improvements in medical treatments and positive changes in cholesterol and blood pressure resulted in a net 29.8% reduction in deaths from 1995 to 2010 [15]. Mexico is

**Table 3. Estimated deaths prevented or postponed by medical or surgical treatments in Mexico 2012.**

| Treatment | No. of elegible patients | Patients receiving treatment | Absolute risk reduction | Deaths prevented or postponed | | | | | |
|---|---|---|---|---|---|---|---|---|---|
| | | | | Best estimate | Minimun estimate | Maximum estimate | Best estimate | Minimun estimate | Maximum estimate |
| | | | | No. Deaths | | | % of total change | | |
| **Myocardial infarction** | | | | | | | | | |
| Aspirin | 17,516 | 0.92 | 0.03 | 47 | -13 | 102 | -0.5% | -1.1% | 0.1% |
| ACE inhibitor | 17,516 | 0.62 | 0.01 | 38 | 27 | 53 | -0.4% | -0.6% | -0.3% |
| Beta blockers | 17,516 | 0.66 | 0.01 | 31 | 21 | 41 | -0.3% | -0.4% | -0.2% |
| CABG | 17,516 | 0.02 | 0.06 | -2 | -7 | 2 | 0.0% | 0.0% | 0.1% |
| PTCA (STEMI) | 7,006 | 0.25 | 0.04 | 30 | 8 | 47 | -0.3% | -0.5% | -0.1% |
| Hospital CPR | 518 | 0.63 | 0.05 | 10 | -613 | 504 | -0.1% | -0.3% | 0.0% |
| Thrombolysis | 17,516 | 0.43 | 0.04 | 81 | -10 | 190 | -0.9% | -2.1% | 0.1% |
| PTCA (NSTEMI) | 10,510 | 0.25 | 0.05 | 48 | -1 | 79 | -0.5% | -0.8% | 0.0% |
| Clopidrogel | 17,516 | 0.90 | 0.00 | 40 | -40 | 25 | -0.4% | 0.4% | -0.3% |
| **Total** | | | | **324** | **212** | **421** | **-3.5%** | **-4.5%** | **-2.3%** |
| **Unstable angina** | | | | | | | | | |
| Aspirin | 13,553 | 0.92 | 0.01 | 13 | -29 | 54 | -0.1% | -0.6% | 0.3% |
| Aspirin & Heparin | 13,553 | 0.59 | 0.02 | 74 | -8 | 173 | -0.8% | -1.8% | 0.1% |
| ACE inhibitor | 13,553 | 0.55 | 0.00 | 10 | -24 | 44 | -0.1% | -0.5% | 0.2% |
| Beta blockers | 13,553 | 0.66 | 0.00 | 14 | -32 | 60 | -0.2% | -0.6% | 0.3% |
| CABG | 13,553 | 0.07 | 0.03 | 5 | 61 | -58 | 0.0% | 0.6% | -0.6% |
| PTCA (STEMI) | 13,553 | 0.38 | 0.02 | 33 | -4 | 85 | -0.4% | -0.9% | 0.0% |
| **Total** | | | | **148** | **-7** | **319** | **-1.6%** | **-3.4%** | **0.1%** |
| **Secondary prevention following myocardial infarction** | | | | | | | | | |
| Statins | 5,073,162 | 0.23 | 0.01 | 424 | 281 | 674 | -4.5% | -7.1% | -2.9% |
| Aspirin | 5,073,162 | 0.75 | 0.00 | 10 | 14 | 6 | -0.1% | -0.1% | -0.2% |
| Warfarin | 5,073,162 | 0.00 | 0.01 | -77 | -216 | 34 | 0.8% | -0.4% | 2.3% |
| ACE inhibitor | 5,073,162 | 0.25 | 0.01 | 218 | -939 | 1102 | -2.3% | -11.7% | 10.1% |
| Beta blockers | 5,073,162 | 0.27 | 0.01 | 148 | 1000 | -830 | -1.6% | 9.0% | -10.7% |
| Rehabilitation | 5,073,162 | 0.03 | 0.01 | 214 | -1096 | 1706 | -2.3% | -18.4% | 11.6% |
| **Total** | | | | **936** | **-2851** | **4474** | **-10.0%** | **-47.8%** | **29.6%** |
| **Secondary prevention following CABG o PTCA** | | | | | | | | | |
| Statins | 118,572 | 0.62 | 0.01 | 80 | 135 | 31 | -0.9% | -0.3% | -1.5% |
| Aspirin | 118,572 | 0.83 | 0.01 | 101 | 261 | -13 | -1.1% | 0.1% | -2.8% |
| Warfarin | 118,572 | 0.00 | 0.01 | -2 | -4 | 0 | 0.0% | 0.0% | 0.0% |
| ACE inhibitor | 118,572 | 0.34 | 0.01 | 63 | 17 | 97 | -0.7% | -1.0% | -0.2% |
| Beta blockers | 118,572 | 0.33 | 0.01 | 42 | 9 | 73 | -0.5% | -0.8% | -0.1% |
| Rehabilitation | 118,572 | 0.04 | 0.01 | 13 | -39 | 73 | -0.1% | -0.8% | 0.4% |
| **Total** | | | | **298** | **-129** | **652** | **-3.2%** | **-7.0%** | **1.4%** |
| **Chronic angina** | | | | | | | | | |
| Statins | 742,011 | 0.30 | 0.01 | 58 | -25 | 141 | -0.6% | -1.5% | 0.3% |
| Aspirin | 742,011 | 0.46 | 0.00 | 108 | 9 | 227 | -1.2% | -2.4% | -0.1% |
| CABG | 742,011 | 0.05 | 0.01 | 348 | 129 | 751 | -3.7% | -8.0% | -1.4% |
| **Total** | | | | **514** | **248** | **845** | **-5.5%** | **-9.1%** | **-2.7%** |
| **Heart failure with hospital admission** | | | | | | | | | |
| Aspirin | 22,539 | 0.44 | 0.05 | 120 | 52 | 200 | -1.3% | -2.1% | -0.5% |
| ACE inhibitor | 22,539 | 0.24 | 0.06 | 82 | 42 | 116 | -0.9% | -1.3% | -0.5% |
| Beta blockers | 22,539 | 0.22 | 0.11 | 118 | 41 | 185 | -1.3% | -2.0% | -0.4% |

*(Continued)*

**Table 3.** (Continued)

| Treatment | No. of elegible patients | Patients receiving treatment | Absolute risk reduction | Deaths prevented or postponed | | | | | |
|---|---|---|---|---|---|---|---|---|---|
| | | | | Best estimate | Minimun estimate | Maximum estimate | Best estimate | Minimun estimate | Maximum estimate |
| | | | | No. Deaths | | | % of total change | | |
| Spironolactone | 22,539 | 0.24 | 0.09 | 130 | 74 | 177 | -1.4% | -1.9% | -0.8% |
| **Total** | | | | **450** | **341** | **588** | **-4.8%** | **-6.3%** | **-3.6%** |
| **Heart failure in the community** | | | | | | | | | |
| Aspirin | 985,831 | 0.60 | 0.01 | 108 | 72 | 147 | -1.2% | -1.6% | -0.8% |
| ACE inhibitor | 985,831 | 0.21 | 0.02 | 163 | -34 | 360 | -1.7% | -3.8% | 0.4% |
| Beta blockers | 985,831 | 0.18 | 0.03 | 140 | 34 | 290 | -1.5% | -3.1% | -0.4% |
| Spironolactone | 985,831 | 0.08 | 0.03 | 90 | 50 | 158 | -1.0% | -1.7% | -0.5% |
| **Total** | | | | **501** | **320** | **714** | **-5.3%** | **-7.8%** | **-3.4%** |
| **Statins for primary prevention** | 9,853,600 | 0.23 | 0.00 | 114 | 68 | 192 | -1.2% | -2.0% | -0.7% |
| **Anti-hypertensive medication** | 11,413,962 | 0.20 | 0.00 | 490 | -148 | 963 | -5.2% | -10.3% | 1.6% |
| **Total treatments** | | | | **3776** | **1744** | **5675** | **-40.3%** | **-61.4%** | **-18.1%** |

CABG indicates coronary artery bypass graft; PTCA Percutaneous transluminal coronary angioplasty; HF heart failure, AH antihypertensive and PCI percutaneous coronary intervention.

one of the few countries in Latin America that still experiences an upward trend for CHD mortality [5, 14]. In the case of countries or regions experiencing upwards trends, IMPACT models were previously implemented in Beijing, Tunisia and Syria; in those cases, risk factors were the main contributors to the increases in CHD mortality [16, 18, 23]. Cholesterol was the primary driver of CHD mortality in Beijing and Tunisia, while blood pressure was the main driver in Syria [16, 18, 23].

In our study, CHD excess mortality was mainly explained through increases in risk factors, mostly changes in diabetes, cholesterol and physical inactivity. These changes occurred along with rapid urbanization and changes in dietary patterns, leading to more physical inactivity and a transition from traditional to Westernised ultra-processed diet [41–45]. These changes in behavioral lifestyles have been associated with an increase in diabetes, obesity and hypercholesterolemia [16, 23, 45, 46]. In our analysis, diabetes was the main contributor to the increase

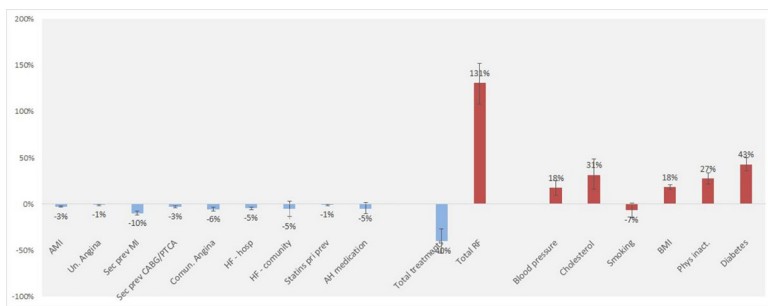

**Fig 1. Proportion of all coronary heart disease deaths explained by the model, which were attributed to the contribution of treatments and risk factors in Mexico, 2000 to 2012.** The bars are the best model estimate and the vertical lines the extreme minimum and maximum estimates in sensitivity analysis. CABG indicates coronary artery bypass graft; PTCA Percutaneous transluminal coronary angioplasty; HF heart failure, AH antihypertensive and PCI percutaneous coronary intervention.

in CHD mortality, given that its self-reported prevalence increased from 5.7% in 2000 to 9.2% in 2012 [47, 48]. The fact that these risk factors share the same fundamental causes, points at the opportunities to implement population-based interventions to provide healthier contexts for diet and physical activity [22]. Over the past decade, Mexico has developed a clear agenda to reduce obesity and metabolic diseases, based on population interventions, such as taxes to unhealthy foods and food warning labels [41, 42]. However, further population-based policies efforts will be needed to reduce obesity, diabetes and CHD deaths in Mexico.

Smoking prevalence fell by 3%, preventing or postponing approximately 670 deaths. However, in the Latin American region, Mexico was one of the first countries to join the Framework Convention on Tobacco Control (FCTC) and has implemented policy changes to reduce tobacco consumption [49]. Main actions included: cigarette taxes increased from 40% in 2002 to 55% of the total price by 2011, national and local smoke-free air laws were implemented, restrictions on tobacco product marketing were strengthened, and prominent pictorial health warnings were required on cigarette packs [49]. However, bigger falls have won large falls in CHD mortality in countries such as the USA, England, and Portugal [2, 3, 19]. Mexico therefore needs to further increase compliance with key tobacco regulations and strengthen the tobacco control regulatory framework to further reduce the smoking prevalence and tobacco-related CHD deaths [49].

Around 2000 and 2012, medical and surgical procedures prevented or postponed nearly 3,900 deaths. The most important contributions came from post-MI secondary treatment, angina treatment and heart failure in the community. Cardiac rehabilitation units in Mexico, increased from 10 in 2009 to 24 in 2015, [50]. However, coverage is still very low, only 4.4% of eligible patients are referred to rehabilitation programs [50]. This is reassuringly consistent with IMPACT model analyses in high-income countries with decreasing CHD mortality [2, 3, 18, 21]. In Syria, the main contribution was from chronic angina treatment [23], from hypertension treatment and myocardial infarction in Beijing [16], and from secondary prevention after MI and hypertension management in Tunisia [18]. This highlights the Rose Principle that the numerically biggest benefits will come from applying effective interventions to the largest patient groups.

Although heart failure therapies in the community had the second-largest contribution to deaths prevented or postponed, previous studies in Mexico suggest that the doses of angiotensin convert enzyme (ACE), spironolactone, and beta-blockers are not optimal [51, 52]. Revascularization from CABG and PTCA together prevented barely 300 deaths, 4% of total CHD deaths, a similar proportion to that observed in Turkey, USA and England, and Wales [2, 3, 16, 24]. Previous studies from the OECD estimated that Mexico has the lowest number of PTCA in the organization [53, 54].

All CHD models have limitations and are dependent on the quality and extent of data available. We made the best efforts to include the most representative and unbiased data available in Mexico. We performed a review to critically summarize the evidence from surveys, registries, and studies that quantified the distribution and frequency of all risk factors included in this model and of most treatment uptakes. Mexican cholesterol data from the National Health and Nutrition Surveys were lacking, we therefore extrapolated information from the Global Burden of Disease (GBD) study [55]. Furthermore, it was not possible to obtain precise data on treatment uptake in Mexico for heart failure and chronic angina treatments. We therefore strengthened our assumptions by obtaining estimates from a consensus group of experts who critically evaluated all the available evidence. The IMPACT model explained 71% of CHD mortality increases; yet, 29% remained unexplained and might reflect data limitations or other unmeasured factors. Finally, we also assumed that the efficiency of therapies in randomized controlled trials could be generalized to population effectiveness in normal clinical practice [3, 22, 36], which could lead to an overestimation of the net benefit of medical interventions.

## Conclusions

Coronary heart disease mortality in Mexico is increasing due to adverse trends in major risk factors and suboptimal use of CHD treatments. Preventive efforts made so far have failed to achieve a substantial impact. Future public policies will therefore need to focus on incentivizing physical activity, strengthening tobacco control policies, promoting healthy foods and discouraging the consumption of processed foods and sugary drinks. Medical and surgical advances have helped to reduce the mortality burden in Mexico; however, their access remains limited and restricted to higher socioeconomic groups. As the country moves to increase coverage for the population [56], an equitable distribution of resources will also be crucial.

## Supporting information

**S1 Appendix. Main data sources for the parameters used in the Mexican IMPACT model for 2000 to 2012.**
(DOCX)

**S2 Appendix. Clinical efficacy of interventions: Relative risk reductions obtained from meta-analyses, and randomized controlled trials.**
(DOCX)

**S3 Appendix. Main assumptions and overlap adjustments used in the Mexican IMPACT model.**
(DOCX)

**S4 Appendix Specific beta coefficients or RR's for major risk factors: Data sources, values and comments.**
(DOCX)

**S5 Appendix. Uncertainty analysis: Parameter distributions, functions and sources.**
(DOCX)

## Author Contributions

**Conceptualization:** Carmen Arroyo-Quiroz, Martin O'Flaherty, Maria Guzman-Castillo, Simon Capewell, Tonatiuh Barrientos-Gutierrez.

**Data curation:** Carmen Arroyo-Quiroz, Eduardo Chuquiure-Valenzuela, Carlos Jerjes-Sanchez.

**Formal analysis:** Carmen Arroyo-Quiroz.

**Investigation:** Tonatiuh Barrientos-Gutierrez.

**Methodology:** Carmen Arroyo-Quiroz, Martin O'Flaherty, Simon Capewell.

**Supervision:** Martin O'Flaherty, Tonatiuh Barrientos-Gutierrez.

**Validation:** Maria Guzman-Castillo.

**Writing – original draft:** Carmen Arroyo-Quiroz.

**Writing – review & editing:** Carmen Arroyo-Quiroz, Martin O'Flaherty, Maria Guzman-Castillo, Simon Capewell, Eduardo Chuquiure-Valenzuela, Carlos Jerjes-Sanchez, Tonatiuh Barrientos-Gutierrez.

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
