## [Decision Letter · Decision Letter 0]

13 Jul 2020

PONE-D-20-12692

Explaining the increment in coronary heart disease mortality in Mexico between 2000 and 2012

PLOS ONE

Dear Dr. Barrientos-Gutierrez,

Thank you for submitting your manuscript to PLOS ONE. After careful consideration, we feel that it has merit but does not fully meet PLOS ONE’s publication criteria as it currently stands. Therefore, we invite you to submit a revised version of the manuscript that addresses the points raised during the review process.

ACADEMIC EDITOR:

In addition to the reviewers' comments, I have the following concerns:

1) Introduction, lines 62-64 can you please elaborate on these issues?

2) Data sources: can you justify the years of data used? The data are not new and thus it would be helpful to state why these data were used.

3) The methods section as presented seems vague and undeveloped. Specifically, the outcome definition lacks details. There is no justification for the variables considered for the attributable risk analysis and how it relates to the outcome. Finally, the statistical analysis is generic and offers little details about what was done to address the aim.

4) Please make sure the tables followed and addressed the aims.

We look forward to receiving your revised manuscript.

Kind regards,

Luisa N. Borrell, DDS, PhD

Academic Editor

PLOS ONE

Journal Requirements:

'Carmen Arroyo-Quiroz received support from CONACyT’s Scholarship Program for

Doctoral Studies. Tonatiuh Barrientos-Gutierrez received support from Harvard University

through the Lown Scholar’s program (https://www.hsph.harvard.edu/lownscholars/scholars/). The funders had no role in the study design or the analysis and interpretation of the data. All authors and their institutions reserve

intellectual freedom from the funders.'

'The authors received no specific funding for this work.'

Reviewers' comments:

Reviewer's Responses to Questions

**Comments to the Author**

1. Is the manuscript technically sound, and do the data support the conclusions?

Reviewer #1: Yes

Reviewer #2: Yes

2. Has the statistical analysis been performed appropriately and rigorously? 

Reviewer #1: Yes

Reviewer #2: Yes

3. Have the authors made all data underlying the findings in their manuscript fully available?

Reviewer #1: Yes

Reviewer #2: Yes

4. Is the manuscript presented in an intelligible fashion and written in standard English?

Reviewer #1: Yes

Reviewer #2: Yes

5. Review Comments to the Author

Reviewer #1: Main comments:

1. The connection between your definition of DPP on page 6 line 113 and the one on page 9 line 172-173 (seem to be observed – expected?) and also your calculation on page 7 line 142 seems vague, please make it more clear and use subscript if necessary (DPP by xx risk factor?).

2. What is the link between DPP and PARF on page 9? Are they always positively related? What is PARF on your table 2 and 3?

3. Gender seems to play an important role for a few risk factors in your table 2, may need to discuss in the texts.

4. Are all the risk factors or treatment considered individually? Have you considered their overlapping part attributable to more than one risk factor?

Minor comments:

A few typos:

1. Page 9 line 178, where is Appendix D?

2. Page 11 line 205, UI 9213, 12,273

3. Table 2 what is Fall in population ?

4. Quite some typos and grammar mistakes, please check carefully

Reviewer #2: Carmen et. al. present an important analysis of the drivers of trends in CHD mortality in Mexico using the validated IMPACT model that has been used in numerous countries. The underlying causes and patterns are different across income strata globally, but seem consistent in terms of growing rates of physical inactivity, obesity, and diabetes. Overall this is a well-written paper and the discussion could be more succint to discuss all the behavioral risk factors in 1 paragraph that likely share underlying fundamental causes (physical inactivity, poor diet leading to greater obesity, HL, + DM). In addition, providing a contrasting or similar viewpoint to other countries across income strata (how is this similar or different in teh trends to US and changing patterns in RF and how does different access to care still contribute to rates).

I have some minor suggestions for the authors' consideration:

1) Avoid colloquial statements or phrasing such as "up-phase" or "the behavior of the CHD epidemic is mixed" and consider describing clearly the trends that differ.

2) Page 6, line 111, should be "observed"

6. PLOS authors have the option to publish the peer review history of their article (what does this mean?). If published, this will include your full peer review and any attached files.

Reviewer #1: No

Reviewer #2: No

---

## [Author Response · Author response to Decision Letter 0]

23 Sep 2020

Response to reviewers

(Please find attached the response to reviewers file)

ACADEMIC EDITOR:

In addition to the reviewers' comments, I have the following concerns:

1) Introduction, lines 62-64 can you please elaborate on these issues?

We have added information about the increased access to CHD treatments, highlighting the importance of surgical and medical treatments for people who was not previously covered under social security (lines 65-69), as follows:

In the past century, Mexico was characterized as a country of low CHD mortality. In 1980, the age-standardized mortality rate per 100,000 inhabitants were 55.6 and 33.8 for men and women, respectively [6]. However, between 1980 and 2010, substantial changes in the country led to a 48% increase in CHD mortality [7]. An explosive increase in obesity and diabetes has been proposed as the main factor for CHD mortality increase [8]. Simultaneously, the country experienced a reduction in smoking rates, from 25.9% in 1980 to 10% in 2012 [9], which should have led to a reduction in coronary morbidity and mortality. Also, since 2004, access to CHD medical and surgical treatments increased under Seguro Popular, a health insurance program to provide access to packages of health services for people without access to social security services [10, 11]. Treatments such as thrombolysis, coronary-artery bypass grafting (CABG), coronary angioplasty, ACE inhibitors and statins are now more commonly used and available for more people [12, 13]. While the rises in CHD mortality in Mexico are clear, we still lack an analysis of the fundamental causes of these changes [14]. 

2) Data sources: can you justify the years of data used? The data are not new and thus it would be helpful to state why these data were used.

We included the following justification in lines 123-126:

In the case National Registry of Acute Coronary Syndromes, the data are available only upon request from the RENASICA Executive Committee, the rest of the datasets are publicly available and anonymous. We limited our CHD mortality analysis to the 2000 to 2012 period, since 2012-2013 was the last wave of RENASICA data.

3) The methods section as presented seems vague and undeveloped. Specifically, the outcome definition lacks details. There is no justification for the variables considered for the attributable risk analysis and how it relates to the outcome. Finally, the statistical analysis is generic and offers little details about what was done to address the aim.

We have re-written this section to make it clearer. The main changes are: 

(i) A detailed explanation of how deaths prevented or postponed (DPPs) were computed and how they are related: 

DPP represents the difference between the 2012 expected CHD deaths, calculated assuming no change in the distribution of risk factors and medical and surgical treatments available in 2000, to the CHD mortality observed in 2012. Mortality rates from CHD were calculated using the underlying cause of death: International Classification of Diseases (ICD) -10 codes I20-I25 [3]. We used demographic data obtained from the Mexican National Population Council (CONAPO) and mortality data for adults aged 25 years and older from the Health Information System from the Mexican Health Ministry to calculate the CHD age- sex-group-specific mortality rates in 2000 and 2012.

We also added information about how DPPtreatment and DPPrisk relate to the main output (DPP): 

The obtained DPP is the number of deaths to be explained by the model, this was achieved thorough the contribution of DPPtreatment ,which represents de DPP that is attributable to changes in medical and surgical treatments, and DPPrisk, that represents the DPP attributable to changes in risk factors 

DPP= Expected mortality 2012 - Observed mortality 2012=DPPtreatment+DPPrisk+e

Where e represents an error term that captures the change that is not explained through the model. In the next sections we will explain how every DPP was calculated.

(ii) We used subcaptions to identify the DPPs from every treatment and risk factor and we added some information at the beginning of all sections to clarify how the calculations relate to the final DPPs. For example, in the case of DPPtreatment:

The first step in the estimation of DPP is to calculate DPPtreatment, which is a combination of the individual DPPtreatment as a result of each intervention/ therapy in each group of patients in 2012, stratified by age and sex. To achieve this, we specified relevant treatments for each of the nine mutually exclusive patient groups [3, 18, 35, 36]:

Then we describe the procedure performed for every patient-treatment group and clarify that: 

This process was replicated for every sex-age group, patient group, and therapy. Some special considerations were made to these initial calculations. We assumed that the proportion of treated patients actually taking therapeutically effective levels of medication (adherence), was 100% among hospitalized patients, 70% among symptomatic patients in the community, and 50% among asymptomatic patients in the community [3, 26].

(iii) We then clarified how the risk factor analysis (PARF section) relates to the main outcome (DPP), lines: 233-245.

We then estimated DPP as the CHD deaths in 2000 multiplied by the difference in the PARF during the period (2000-2012).

DPPrisk=CHDdeaths2000*(PARF2012-PARF2000)

For example, suppose that the prevalence of diabetes among men aged 65-74 years was 14.5% in 2000 and 20.7% in 2012. Assuming a RR= 1.93, the PARF in 2000 was 0.119 and 0.161 in 2010. The number of CHD deaths is 2000 was 123,055. The DPP attributable to the change in diabetes prevalence was therefore: 

DPPdiabetes= (123,055) * (0.161 - 0.119) = 5,168 

This calculation was then repeated for each sex-age group and for every categorial risk factor. Finally, we obtained a total DPPrisk for each sex-age group by adding the DPP obtained for each risk factor across all risk factors.

4) Please make sure the tables followed and addressed the aims.

Done

Reviewer #1: Main comments:

1. The connection between your definition of DPP on page 6 line 113 and the one on page 9 line 172-173 (seem to be observed – expected?) and also your calculation on page 7 line 142 seems vague, please make it more clear and use subscript if necessary (DPP by xx risk factor?).

We re-wrote the methods section to clarify this part and we decided, as the reviewer suggested, to use subscripts to identify DPPs. Briefly, the original text stated: 

DPP represents the difference between the 2012 expected CHD deaths, calculated assuming no change in the distribution of risk factors and medical and surgical treatments available in 2000, to the CHD mortality observed in 2012.

DPP= Expected mortality 2012 - Observed mortality 2012

And we added information to relate every DPP calculated to the original DPP (lines 143-149): 

The obtained DPP is the number of deaths to be explained by the model, this was achieved thorough the contribution of DPPtreatment,which represents de DPP from treatments, and DPPrisk, that represents the DPP from risk factors. In the next sections we will explain how every DPP was calculated.

DPP= Expected mortality 2012 - Observed mortality 2012=DPPtreatment+DPPrisk+e

Where e represents an error term or the part that is not explained through the model. 

Finally, in the Medical treatment section and in the Risk factors sections we describe how the DPP for every risk factor o medical treatment were going to be calculated to contribute to the final result. 

2. What is the link between DPP and PARF on page 9? Are they always positively related? What is PARF on your table 2 and 3?

We added some information in the methodology section to clarify this part (lines 235-247). 

We then estimated DPP as the CHD deaths in 2000 multiplied by the difference in the PARF during the period (2000-2012).

DPPrisk=CHDdeaths2000*(PARF2012-PARF2000)

For example, suppose that the prevalence of diabetes among men aged 65-74 years was 14.5% in 2000 and 20.7% in 2012. Assuming a RR= 1.93, the PARF in 2000 was 0.119 and 0.161 in 2010. The number of CHD deaths is 2000 was 123055. The DPP attributable to the change in diabetes prevalence was therefore: 

DPPdiabetes= (123055) * (0.161 - 0.119) = 5168 

This calculation was then repeated for each sex-age group and for every categorial risk factor. Finally, we obtained an unique DPPrisk per each sex-age group combining the DPP obtained for each risk factor studied.

They are not always positively related, it depends on the result of the expected-observed difference. In tables 2 and 3 we include the DPPrisk that is related to all risk factors analyzed.

3. Gender seems to play an important role for a few risk factors in your table 2, may need to discuss in the texts.

 We included the following lines (286-290): 

Some risk differences between men and women are important; women had larger increases in diabetes, smoking and cholesterol, while men had larger increases in systolic blood pressure and physical inactivity. Although women experienced a larger increase in smoking prevalence, the prevalence for men remained higher. 

4. Are all the risk factors or treatment considered individually? Have you considered their overlapping part attributable to more than one risk factor?

In the methodology section lines 191-194 we described how we managed the overlapping in the case of patient treatments. 

Potential overlaps between different groups of patients were detected and adjustments were made to prevent double-counting (e.g., 50% of patients having CABG surgery had previous myocardial infarction) (S3 Appendix) [3, 18, 26]. Briefly, we subtracted the DPPs calculated in the treatment component from the DPPs calculated in the risk factors component.

In the case of risk factors, in lines 247-250 we commented the assumptions we made as follows: 

As independent coefficients of regression and relative risks were derived from multivariate analyses for each risk factor, we assumed that there was no further overlap across all risk factors considered

Minor comments:

A few typos:

1. Page 9 line 178, where is Appendix D?

We have corrected it; we referred to supplementary material S4 Appendix.

2. Page 11 line 205, UI 9213, 12,273

Those numbers were negative, we have included the corresponding sign. The correct interval is: 

(UI -12,273; -9,213 Table 2).

3. Table 2 what is Fall in population?

It was used to refer to the change of that risk factor in the population but, to be consistent with the rest of the table, we change it to ALL to indicate that it combines changes for women and men.

4. Quite some typos and grammar mistakes, please check carefully

Solved

Reviewer #2: 

Carmen et. al. present an important analysis of the drivers of trends in CHD mortality in Mexico using the validated IMPACT model that has been used in numerous countries. The underlying causes and patterns are different across income strata globally, but seem consistent in terms of growing rates of physical inactivity, obesity, and diabetes. Overall this is a well-written paper and the discussion could be more succint to discuss all the behavioral risk factors in 1 paragraph that likely share underlying fundamental causes (physical inactivity, poor diet leading to greater obesity, HL, + DM). In addition, providing a contrasting or similar viewpoint to other countries across income strata (how is this similar or different in teh trends to US and changing patterns in RF and how does different access to care still contribute to rates).

We changed the discussion to summarize the information about risk factors in a single paragraph.(lines 353-366)

In our study, CHD excess mortality was mainly explained through increases in risk factors, mostly changes in diabetes, cholesterol and physical inactivity. These changes occurred along with rapid urbanization and changes in dietary patterns, leading to more physical inactivity and a transition from traditional to Westernised ultra-processed diet [42-45]. These changes in behavioral lifestyles have been associated with an increase in diabetes, obesity and hypercholesterolemia [44, 49, 51, 52]. In our analysis, diabetes was the main contributor to the increase in CHD mortality, given that its self-reported prevalence increased from 5.7% in 2000 to 9.2% in 2012 [53, 54]. The fact that these risk factors share the same fundamental causes, points at the opportunities to implement population-based interventions to provide healthier contexts for diet and physical activity [24]. In recent, Mexico has developed a clear agenda to reduce obesity and metabolic diseases, based on population interventions, such as taxes to unhealthy foods and food warning labels [43, 44]. However, further population-based policies efforts will be needed to reduce obesity, diabetes and CHD deaths in Mexico.

We added some information to compare with other countries in the discussion section lines 339-352.

Most of the previous IMPACT models were implemented in high income countries that are experiencing a decrease in CHD mortality, mostly attributable to risk factor reductions [3, 18, 21, 37, 41]. Those are the cases of United Kingdom, Denmark , Japan, Netherlands or the United States, among others [3, 21, 26, 36, 41]. In Latin America, only Argentina has implemented the IMPACT model and, while they have experienced some increases in diabetes and obesity, the improvements in medical treatments and positive changes in cholesterol and blood pressure resulted in a net 29.8% reduction in deaths from 1995 to 2010 [16]. Mexico is one of the few countries in Latin America that still experiences an upward trend for CHD mortality [5, 14]. In the case of countries or regions experiencing upwards trends, IMPACT models were previously implemented in Beijing, Tunisia and Syria; in those cases, risk factors were the main contributors to the increases in CHD mortality [17, 19, 24]. Cholesterol was the primary driver of CHD mortality in Beijing and Tunisia, while blood pressure was the main driver in Syria [17, 19, 24]. 

I have some minor suggestions for the authors' consideration:

1) Avoid colloquial statements or phrasing such as "up-phase" or "the behavior of the CHD epidemic is mixed" and consider describing clearly the trends that differ.

We changed “up-phase” for “growing-phase” in line 26

We changed “epidemic is mixed” for “epidemic differs” in line 52

2) Page 6, line 111, should be "observed"

Done

---

## [Decision Letter · Decision Letter 1]

12 Nov 2020

Explaining the increment in coronary heart disease mortality in Mexico between 2000 and 2012

PONE-D-20-12692R1

Dear Dr. Barrientos-Gutierrez,

We’re pleased to inform you that your manuscript has been judged scientifically suitable for publication and will be formally accepted for publication once it meets all outstanding technical requirements.

Kind regards,

Luisa N. Borrell, DDS, PhD

Academic Editor

PLOS ONE

Additional Editor Comments (optional):

Reviewers' comments:

Reviewer's Responses to Questions

**Comments to the Author**

1. If the authors have adequately addressed your comments raised in a previous round of review and you feel that this manuscript is now acceptable for publication, you may indicate that here to bypass the “Comments to the Author” section, enter your conflict of interest statement in the “Confidential to Editor” section, and submit your "Accept" recommendation.

Reviewer #1: All comments have been addressed

Reviewer #2: All comments have been addressed

2. Is the manuscript technically sound, and do the data support the conclusions?

Reviewer #1: Yes

Reviewer #2: Yes

3. Has the statistical analysis been performed appropriately and rigorously? 

Reviewer #1: Yes

Reviewer #2: Yes

4. Have the authors made all data underlying the findings in their manuscript fully available?

Reviewer #1: Yes

Reviewer #2: Yes

5. Is the manuscript presented in an intelligible fashion and written in standard English?

Reviewer #1: Yes

Reviewer #2: Yes

6. Review Comments to the Author

Reviewer #1: (No Response)

Reviewer #2: I have no further comments. The authors report on an important topic in a clear and excellent manner.

7. PLOS authors have the option to publish the peer review history of their article (what does this mean?). If published, this will include your full peer review and any attached files.

Reviewer #1: No

Reviewer #2: No

---

## [Editor Report · Acceptance letter]

18 Nov 2020

PONE-D-20-12692R1 

Explaining the increment in coronary heart disease mortality in Mexico between 2000 and 2012 

Dear Dr. Barrientos-Gutierrez:

I'm pleased to inform you that your manuscript has been deemed suitable for publication in PLOS ONE. Congratulations! Your manuscript is now with our production department. 

Kind regards, 

on behalf of

Dr. Luisa N. Borrell 

Academic Editor

PLOS ONE